# Co-Occurrence of Free-Living Amoeba and *Legionella* in Drinking Water Supply Systems

**DOI:** 10.3390/medicina55080492

**Published:** 2019-08-15

**Authors:** Olga Valciņa, Daina Pūle, Artjoms Mališevs, Jūlija Trofimova, Svetlana Makarova, Genadijs Konvisers, Aivars Bērziņš, Angelika Krūmiņa

**Affiliations:** 1Institute of Food Safety, Animal Health and Environment “BIOR”, LV-1076 Rīga, Latvia; 2Department of Water Engineering and Technology, Riga Technical University, LV-1658 Rīga, Latvia; 3Department of Infectology and Dermatology, Riga Stradiņš University, LV-1007 Rīga, Latvia

**Keywords:** *Legionella*, amoeba, co-occurrence, water, protozoa

## Abstract

*Background and Objectives:**Legionella* is one of the most important water-related pathogens. Inside the water supply systems and the biofilms, *Legionella* interact with other bacteria and free-living amoeba (FLA). Several amoebas may serve as hosts for bacteria in aquatic systems. This study aimed to investigate the co-occurrence of *Legionella* spp. and FLA in drinking water supply systems. *Materials and Methods:* A total of 268 water samples were collected from apartment buildings, hotels, and public buildings. Detection of *Legionella* spp. was performed in accordance with ISO 11731:2017 standard. Three different polymerase chain reaction (PCR) protocols were used to identify FLA. *Results:* Occurrence of *Legionella* varied from an average of 12.5% in cold water samples with the most frequent occurrence observed in hot water, in areas receiving untreated groundwater, where 54.0% of the samples were *Legionella* positive. The occurrence of FLA was significantly higher. On average, 77.2% of samples contained at least one genus of FLA and, depending on the type of sample, the occurrence of FLA could reach 95%. In the samples collected during the study, *Legionella* was always isolated along with FLA, no samples containing *Legionella* in the absence of FLA were observed. *Conclusions:* The data obtained in our study can help to focus on the extensive distribution, close interaction, and long-term persistence of *Legionella* and FLA. Lack of *Legionella* risk management plans and control procedures may promote further spread of *Legionella* in water supply systems. In addition, the high incidence of *Legionella*-related FLA suggests that traditional monitoring methods may not be sufficient for *Legionella* control.

## 1. Introduction

*Legionella* is gram-negative, ubiquitous bacteria that are most commonly found in man-made water systems. *Legionella* causes Legionnaires’ disease, an acute pneumonia-like infection with high case-fatality rate and Pontiac fever, an influenza-like, self-limited illness that is not notifiable in the majority of countries and therefore, underestimated [1,2].

*Legionella* is one of the most important water-related pathogens, which can lead to both outbreaks and sporadic cases. Dissemination of *Legionella* is carried out through water aerosols containing *Legionella*, which occur in water supply systems, such as shower handsets, taps, cooling towers, fountains, and spas. In the form of aerosols, bacteria can be spread kilometers away from the primary source of infection while still maintaining viability [3].

Inside the water supply systems and the biofilms, *Legionella* come into contact with other bacteria and free-living amoeba (FLA). The most frequently isolated FLA are *Acanthamoeba*, *Vermaoeba,* and *Naegleria* [4]. Certain amoeba genotypes, such as *Acanthamoeba* T4 and *Naegleria fowleri,* can be human and animal pathogens. *Acanthamoeba* can cause amoebic keratitis, which can lead to blindness if not treated, while brain infections can cause fatal Granulomatous Amoebic Encephalitis. *N. fowleri* can cause primary amoebic meningoencephalitis, which is often fatal [5].

Despite the fact that the majority of amoebas are predators of other microorganisms, especially bacteria, interaction between them has been observed. Several amoeba species may serve as hosts for bacteria in a natural environment and man-made water systems, such as water pipes, cooling towers, etc., as bacteria have developed adaptation mechanisms over time to escape predating [6,7]. As a result, bacteria may not only be amoeba resistant, they may be able to multiply in the amoebas and take advantage of the benefits they can offer, such as a nutrient source and additional protection against temperature, disinfectants or UV radiation [8,9]. Mechanisms that have helped bacteria escape predation can become factors of bacterial virulence [10,11,12]. Parasitizing FLA and being a member of mixed species biofilm increases the capacity to produce polysaccharides and to form biofilms [13,14].

In Latvia, an average of 1.5 cases of Legionnaires’ disease are reported per 100,000 inhabitants, which is very similar to the EU average [15]. However, the average age of residential buildings in Latvia is high, and only a small part of the buildings have renovated water supply systems [16]. There are no previously published studies on the diversity of FLA and co-occurrence with *Legionella* in water supply systems of residential and other buildings in Latvia. This study aimed to investigate the co-occurrence of *Legionella* spp. and FLA in drinking water supply systems to understand the interaction between *Legionella* and FLA and potential consequences to public health concerns.

## 2. Materials and Methods

### 2.1. Sampling

A total of 268 water samples, one liter each, were collected, including 44 cold water samples and 224 hot water samples from apartment buildings, hotels, and public buildings (sports clubs, offices, etc.) from August 2017 until May 2018. Buildings supplied with drinking water from different water sources (i.e., groundwater and treated surface water) were included in the sampling plan. Samples were taken in Riga (139 of 268 samples) and other cities (129 of 268 samples) in Latvia (Figure 1 and Figure 2). Overall, 92 buildings were included in the sampling plan, 42 of them in Riga and 50 in other cities in Latvia. The temperature of the water was measured after the sampling and after at least five minutes of flushing.

### 2.2. Detection of Legionella and Free-Living Amoeba

One liter water samples were filtered through membrane filters of 47 mm diameter and pore size 0.45 µm (Mixed Cellulose Ester, Membrane Solutions, Kent, WA, USA). Membrane filters were cut in small pieces and put in Petri plate with 5 mL of distilled water, as required for *Legionella* analyses. Filter pieces and the remaining suspensions were used for amoeba cultivation. Detection of *Legionella* spp. was performed in accordance with ISO 11731:2017 standard [17]. A total of three 0.1 mL untreated, heat-treated, and acid-treated aliquots of the samples were spread on Buffered Charcoal Yeast extract medium (GVPC, Oxoid, Hampshire, UK). The plates were incubated at 36 °C in a humidified environment for 10 days and examined every day starting on day 3. At least three characteristic colonies from each GVPC plate were selected for subculture onto plates Buffered Charcoal Extract agar medium with L-cysteine (BCYE, Oxoid, Hampshire, UK) and Buffered Charcoal Extract agar medium without L-cysteine (BCYE-Cys, Oxoid, Hampshire, UK) and incubated for at least 48 h at 36 °C.

For cultivation of amoeba, Page’s Amoeba Saline (PAS) was used. PAS media was prepared from two stocks: the first contained NaCl—12.0 g, MgSO_4_ × 7H_2_O—0.40 g, CaCl_2_ × 2H_2_O—0.60 g per 500 mL of water, the second: Na_2_HPO_4_—14.20 g, KH_2_PO_4_—13.60 g per 500 mL of water. For the preparation of the final PAS media, 500 mL of each stock was used. Two sterilized rice grains with 15 mL of final PAS media were added to the Petri plate with resuspended membrane filter pieces. Plates were incubated four to five days at temperature +25 °C. After incubation Petri plates were examined under the light microscope using keys [18,19]. For enrichment of amoebas before DNA extraction 70 µL of liquid Peptone–yeast–glucose (PYG) medium (proteose peptone 20 g, glucose 18 g, yeast extract 2 g, sodium citrate dihydrate 1 g, MgSO_4_ × 7H_2_O 0.98 g, Na_2_HPO_4_ × 7H_2_O 0.355 g, KH_2_PO_4_ 0.34 g, Fe(NH_4_)_2_(SO_4_)_2_ × 6H_2_O 0.02 g, distilled water 1000 mL) was used.

### 2.3. Identification of Free-Living Amoeba

DNA was extracted with a Flexi Gene DNA kit (QIAGEN, Hilden, Germany), for *Acanthamoeba* detection, polymerase chain reaction (PCR) was done using a protocol previously described [20]. Amplicon ASA.S1 (423- to 551-bp) *Acanthamoeba*-specific amplimer from 18S rDNA that were highly specific for the genus *Acanthamoeba* was used. It is obtained with primers JDP1 and JDP2 [20]), containing DF3 (diagnostic fragment 3 represents a region of the gene that is also within the amplimer ASA.S1. DF3 can be analyzed with other primers), and was amplified by PCR using genus-specific primers JDP1and JDP2. *Amoebaidae* and *Vahlkampfidae* detection was done using a protocol previously described [21]. Two amplification protocols were applied: For amplification of the 150 bp fragment for *Vahlkampfidae* DNA, primers Vahl_560F and Vahl_730R were used. The 18S rRNA gene from FLA was amplified by PCR using primers Amo_1400F and Amo_1540R which amplify a 130 bp fragment for *Acanthamoeba* DNA and a 50 bp fragment for *Echinamoeba* and *Vermamoeba* DNA. For representatives of former Genus *Hartmanella* detection, PCR was done using a protocol previously described [22]. The 18S rRNA gene from FLA was amplified by PCR using primers Hartm F and Hartm R. PCR products were prepared for sequencing by purification using USB ExoSAP-IT PCR product clean-up (Affymetrix, Inc., Santa Clara, CA, USA). The Big Dye Terminator v3.1 kit was used (ThermoFisher Scientific, Waltham, MA, USA) according to the manufacturer’s protocol. Homology analysis of the obtained sequences with genes in the gene data bank was done using BLAST software from the National Center for Biotechnology Information (NCBI) site. Strains were identified by the highest homology and query coverage.

### 2.4. Data Analysis

Statistical analysis was performed by SPSS Statistics Version 22 (IBM Corporation, Chicago, IL, USA). Contingency tables, Chi-squared tests, and correlation analysis were used to evaluate the association between the occurrence of *Legionella* spp. and FLA and various factors. Maps were created by QGIS version 3.4.6.

## 3. Results

Overall, *Legionella* spp. were detected in 114 of 268 water samples (Table 1), including 11 of 44 cold water samples (25.0%) and 103 of 224 hot water samples (46.0%). At least one *Legionella* spp. positive sample was detected in 50 of 92 buildings (54.3%).

The most commonly observed *Legionella* species were *Legionella pneumophila*, which was detected in 105 from 114 *Legionella* spp. positive samples (92.1%). *Legionella rubrilucens* was observed in seven samples (6.1%), and *Legionella anisa* was found in two samples (1.8%). The most predominant *L. pneumophila* serogroup (sg) was sg 3, which was identified in 60 of 105 cases (57.1%). *L. pneumophila* sg 2 was identified in 17 cases (16.2%) and sg 1 in 16 cases (15.2%), while serogroups 6 and 9 were observed in six cases each.

Level of *Legionella* spp. colonization ranged from 50 cfu/L to 13 × 10^3^ cfu/L. For cold water samples, the average level of contamination was 4.6 × 10^3^ cfu/L (min 100 cfu/L, max 12 × 10^3^ cfu/L) and 1.5 × 10^3^ cfu/L for hot water samples (min 5 cfu/L, max 13 × 10^3^ cfu/L). Overall, higher levels of *Legionella* colonization were observed in samples from apartment buildings (average 3.3 × 10^3^ cfu/L, min 50 cfu/L, max 13 × 10^3^ cfu/L) than in public buildings (average 1.4 × 10^3^ cfu/L, min 50 cfu/L, max 6.5 × 10^3^ cfu/L), and hotels (average 8.9 × 10^2^ cfu/L, min 5 cfu/L, max 7.5 × 10^3^ cfu/L).

Chi-squared tests showed no association between water source (χ^2^ = 1.6, *p* = 0.25), type of building (χ^2^ = 1.1, *p* = 0.57) and occurrence of *Legionella* spp. No differences in the occurrence of *Legionella* spp. were observed between sampling season as well (χ^2^ = 0.5, *p* = 0.50). Significantly higher occurrence of *Legionella* spp. was observed in hot water samples (χ^2^ = 6.6, *p* = 0.01). The presence of *Legionella* spp. was significantly higher in hot water samples with temperature less than 50 °C (χ^2^ = 7.6, *p* = 0.049), while the temperature of the cold water did not show any impact on the occurrence of *Legionella* spp.

The average temperature of the cold water was 13.1 ± 0.9 °C. Temperature measurements showed that in five cases, the temperature of the cold water was equal to or above 20 °C.

The average temperature of the hot water was 46.6 ± 0.7 °C. Temperature measurements showed that only in 38 of 224 hot water samples’ temperature exceeded 55 °C (Table 2).

No significant differences were observed between temperatures in apartment buildings, public buildings, and hotels in Riga and other cities (*p* = 0.43).

FLA were observed in 207 of 268 water samples (Table 3), including 37 cold water samples (84.1%) and in 170 hot water samples (75.9%). At least one FLA positive sample was observed in 83 of 92 buildings (90.2%).

Chi-squared tests showed no association between water type (i.e., cold or hot) and the presence of FLA (χ^2^ = 1.4, *p* = 0.33). However, a higher diversity of FLA was observed in hot water samples (χ^2^ = 10.3, *p* = 0.0.35).

The presence of FLA was significantly higher in hot water samples with temperature less than 50 °C (χ^2^ = 21.3, *p* < 0.0001), while the temperature of the cold water did not show any impact on the occurrence of FLA.

The sampling season and type of building showed no association with presence of FLA (χ^2^ = 6.0, *p* = 0.11 and χ^2^ = 3.9, *p* = 0.14, respectively). A higher occurrence of FLA was observed in samples from buildings which received groundwater (χ^2^ = 5.8, *p* = 0.024). However, the water source had no impact on the diversity of FLA (χ^2^ = 7.5, *p* = 0.11).

Overall eight genus of FLA were identified in 207 samples, including *Acanthamoeba* spp. (146 samples), *Vermamoeba* spp. (77), *Naegleria* spp. (58), *Flamella* spp. (3), *Centropyxis* spp. (2), *Vrihiamoeba* spp. (2), *Echinamoeba* spp. (1), and *Tetramitus* spp. (1).

In 47.4% of samples, only one genus of amoeba was observed (127 of 268 cases). In 69 samples (25.7%), two different genera were detected, while in 10 cases (3.7%) there were three genera and in one sample (0.4%) four different amoeba species were detected. Most frequently observed combinations of amoeba species are shown in Table 4.

Association between the presence of FLA and the occurrence of *Legionella* spp. was also observed (χ^2^ = 58.5, *p* < 0.0001). No *Legionella* spp. positive samples were observed in the absence of FLA (Table 5) and the concordance rate between the presence of FLA and *Legionella* spp. reached 55.1%. Correlation analysis revealed a moderate positive correlation (r = 0.467, *p* < 0.01) between the presence of *Legionella* and FLA in water samples.

The occurrence of *Legionella* spp. was significantly higher in samples with lower diversity of FLA (χ^2^ = 64.9, *p* < 0.0001). The highest occurrence of *Legionella* spp. was observed in samples with only one amoeba genera (Table 6).

The diversity of FLA did not show any impact on *Legionella* species (χ^2^ = 1.9, *p* = 0.758), serogroups (χ^2^ = 4.6, *p* = 0.797) or on count of colony forming units (χ^2^ = 7.9, *p* = 0.247).

Only one amoeba genus showed significant association with the occurrence of *Legionella* spp.—*Acanthamoeba* spp., which was observed in 146 samples. Correlation analysis revealed moderate positive correlation (r = 0.404, *p* < 0.001) between the presence of *Acanthamoeba* spp. and *Legionella* spp. in the samples. *Vermamoeba* spp., which was observed in 77 samples (r = 0.131, *p* = 0.074), and other genus did not show a statistically significant impact on the occurrence of *Legionella* spp.

Overall, 14 species of FLA were identified. Only two genus—*Acanthamoeba* spp. and *Naegleria* spp. contained more than one species. From identified species, the most common were *Vermamoeba* vermiformis (54%), *Acanthamoeba* castellanii (12%) and *Acanthamoeba* polyphaga (8%).

## 4. Discussion

Our research showed that FLA and *Legionella* are common in drinking water in Latvia and water supply systems in multi-apartment buildings, hotels, sports clubs, and office buildings can become a potential source of *Legionella* infection.

Contamination of drinking water with *Legionella* was relatively high (42%). The occurrence of *Legionella* varied from an average of 12.5% in cold water samples with the most frequent occurrence observed in hot water, in areas receiving untreated groundwater, where 54% of the samples were *Legionella* positive. FLA and *Legionella* occurrence and diversity did not differ significantly for samples taken within one building. Similar studies in Japan revealed *Legionella* prevalence in 6.5% of water samples using a culture method [23], while in the United Kingdom, *Legionella* was found in 8% of water samples taken in household showers [24]. In both studies, *Legionella* was also analyzed using a second method—loop-mediated isothermal amplification in Japan and quantitative PCR in the United Kingdom, and the observed incidence of *Legionella* changed to 47.8% and 31%, respectively. Given the nature of DNA-based techniques, it is possible that in our study, the incidence of *Legionella* would be even higher if additional molecular techniques were used. However, the occurrence of *Legionella* may vary significantly from country to country. Prevalence of *Legionella* in potable water in Germany varied from 20.7% [25] to 32.7% [26], in Italian residential buildings different *Legionella* species were detected in 26% of the hot water networks [27].

In this study, close similarities with the study carried out in Hungary [28] were observed. Between the years 2006 and 2013, a total of 1809 water samples were taken in Hungary in 168 different buildings, and 60% of buildings were colonized by *Legionella*, 46% of hot water samples were positive for *Legionella* [28]. The main reasons for the high contamination are also similar in both countries. The Hungarian authors mention the low temperature of hot water and the lack of adequate monitoring and risk management. In our study, the average temperature of hot water was 46.6 °C, which is appropriate for maintaining the viability of *Legionella*, with only 17% of samples of hot water above 55 °C, which would be the preferred temperature to avoid the proliferation of *Legionella* [29].

The low temperature in water supply systems can have several explanations. First, it relates to the overall economic situation and the public’s understanding of energy saving. Second, particularly in public buildings and hotels, it may be linked to a lack of staff competence, and third, in a large part of old buildings without the complete renovation of the water supply system, there is no technical solution to raise the temperature, since the old facilities are not suitable for maintaining temperatures above 55 °C. However, raising temperatures alone will not lead to significant improvements without the necessary dimensioning and recirculation [30].

The occurrence of FLA was significantly higher. On average, 77.2% of samples contained at least one genus of FLA and, depending on the type of sample, the occurrence of FLA could reach 95%.

The most commonly identified were *Acanthamoeba* (54.5% of all) and *Vermamoeba* (28.7% of all), followed by vahlkampfiid amoebae, yet more than 20% of the samples contained more than one genus of amoebas, most commonly *Acanthamoeba* and *Vermamoeba* were found together. Other studies have also confirmed that FLA was found in drinking water and environmental samples [31,32,33,34,35], biofilms [36,37], in industrial waters [38], and cooling towers [39].

In the samples collected during this study, *Legionella* was always isolated along with FLA, no samples containing *Legionella* in the absence of FLA were observed. Co-occurrence of *Legionella* and FLA in the water systems may indicate an increased health risk in proximal areas of the system, where lower temperatures are commonly observed [8]. In addition to protecting *Legionella*, FLA are also able to maintain the viable but non-culturable strains [40] and provide long-term persistence and transmission of *Legionella* [41].

Although Legionnaires disease is included in the list of notifiable diseases in Latvia, currently there are no regulations in Latvia for the introduction of risk management plans and regular environmental monitoring of *Legionella*, and, in most cases, the risk of *Legionella* is not taken into account in the management of buildings. Thus, the minimum requirements for hot water temperature at the points of consumption are not regulated in Latvia. Many countries have guidelines and regulations for the prevention of the Legionnaires disease. However, the regulatory framework for *Legionella* control differs between countries [42]. Most of the guidelines and regulations are focused on *Legionella* control, with no regard to the presence of FLA. The current approach in *Legionella* control involves monitoring of *Legionella* sp. with culture-based methods and does not allow to assess true concentration of *Legionella* cells [43]. The data obtained in our study can help to focus on the extensive distribution, close interaction, and long-term persistence of *Legionella* and FLA. However, further studies may be needed to identify whether FLA-targeted risk management plans will contribute to decreasing the risk to public health.

## 5. Conclusions

Lack of *Legionella* risk management plans and control procedures may promote further spread of *Legionella* in water supply systems. In addition, a high incidence of *Legionella*-related FLA suggests that traditional monitoring methods may not be sufficient for *Legionella* control.

## Figures and Tables

**Figure 1 medicina-55-00492-f001:**
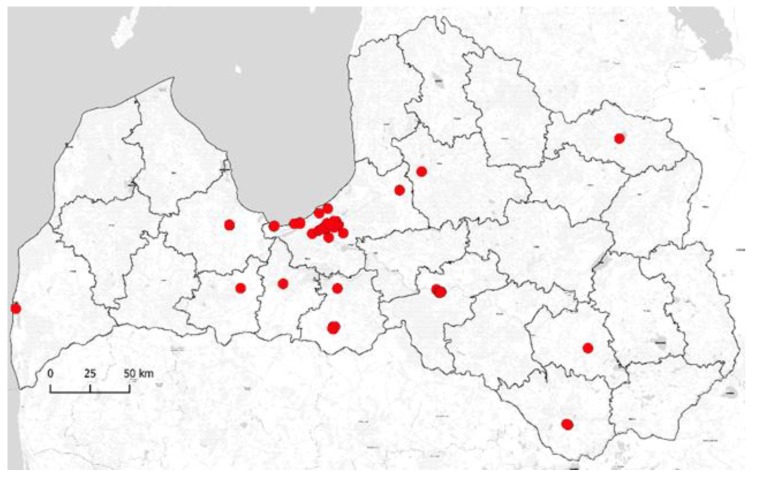
Geographical distribution of sampling points from water supply systems in Latvia.

**Figure 2 medicina-55-00492-f002:**
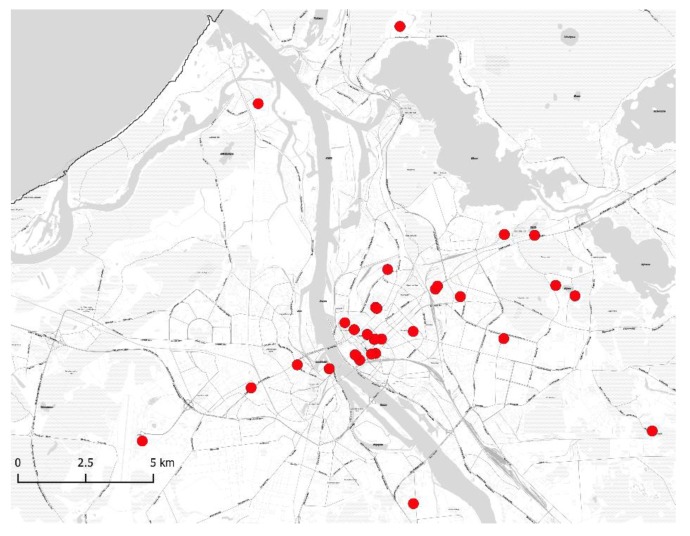
Geographical distribution of sampling points from water supply systems in Riga.

**Table 1 medicina-55-00492-t001:** Number of tested samples and *Legionella* spp. positive samples.

	Water Source/Samples Tested (Positive; %)				
	Treated Surface Water		Groundwater		Total
	Cold Water	Hot Water	Cold Water	Hot Water	
Apartment buildings	4 (1; 25.0%)	13 (3; 23.1%)	24 (6; 25.0%)	41 (22; 53.7%)	82 (32; 39.0%)
Hotel	0 (0; 0.0%)	75 (33; 44.0%)	0 (0; 0.0%)	68 (32; 47.1%)	143 (65; 45.5%)
Public buildings	4 (0; 0.0%)	5 (1; 20.0%)	12 (4; 33.3%)	22 (12; 54.5%)	43 (17; 39.5%)
Subtotal	8 (1; 12.5%)	93 (37; 39.8%)	36 (10; 27.8%)	131 (66; 50.4%)	268 (114; 42.5%)
Total	101 (38; 37.6%)		167 (76; 45.5%)		

**Table 2 medicina-55-00492-t002:** Occurrence of *Legionella* spp. in water samples at different temperature ranges.

Type of Building	*Legionella* Positive Hot Water Samples	
	Temperature Below 55 °C	Temperature Above or Equal to 55 °C
Apartment buildings	13 of 26 (50.0%)	10 of 23 (43.5%)
Hotels	62 of 128 (48.4%)	2 of 14 (14.3%)
Public buildings	4 of 15 (26.7%)	0 of 1 (0.0%)
Total	79 of 169 (46.7%)	12 of 38 (31.6%)

**Table 3 medicina-55-00492-t003:** Number of PCR tested samples and FLA positive samples in water supply systems.

	Water Source/Samples Tested (Positive; %)				
	Treated Surface Water		Groundwater		Total
	Cold Water	Hot Water	Cold Water	Hot Water	
Apartment buildings	4 (2; 50.0%)	13 (6; 46.1%)	24 (21; 87.5%)	41 (31; 75.6%)	82 (60; 73.2%)
Hotel	0 (0; 0.0%)	75 (55; 73.3%)	0 (0; 0.0%)	68 (54; 79.4%)	143 (109; 76.2%)
Public buildings	4 (4; 100.0%)	5 (3; 60.0%)	12 (10; 83.3%)	22 (21; 95.4%)	43 (38; 88.4%)
Subtotal	8 (6; 75.0%)	93 (64; 68.8%)	36 (31; 86.1%)	131 (106; 80.9%)	268 (207; 77.2%)
Total	101 (70; 69.3%)		167 (137; 82.0%)		

**Table 4 medicina-55-00492-t004:** Most frequently observed amoebal families (%).

Amoebal Family	Number of Samples (%)
*Acanthamoebidae*	74 (27.6%)
*Vahlkampfiidae*	29 (10.8%)
*Vermamoebidae*	24 (9.0%)
*Acanthamoebidae + Vermamoebidae*	43 (16.0%)
*Acanthamoebidae + Vahlkampfiidae*	15 (5.6%)

**Table 5 medicina-55-00492-t005:** Co-occurrence of FLA and *Legionella* spp. in water supply systems.

	*Legionella* spp. Negative Samples (%)	*Legionella* spp. Positive Samples (%)
FLA negative samples (%)	61 (100.0%)	0 (0.0%)
FLA positive samples (%)	93 (44.9%)	114 (55.1%)

**Table 6 medicina-55-00492-t006:** Occurrence of *Legionella* spp. in water samples with different FLA genus count.

Number of Amoeba Genus	*Legionella* spp.	
	Negative (%)	Positive (%)
1	49 (38.6%)	78 (61.4%)
2	37 (53.6)	32 (46.4%)
3	6 (60.0)	4 (40.0)
4	1 (100.0)	0 (0.0)

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
