# Peer review of "Co-Occurrence of Free-Living Amoeba and *Legionella* in Drinking Water Supply Systems"

_medicina, 2019, doi:10.3390/medicina55080492_

Round 1

Reviewer 1 Report

General Comments

A report confirming data that Legionella and free-living amoebae co-inhabit drinking water systems.

Lacking is proof that legionella must have amoebae for their persistence in drinking water.

Specific Comments

Line 14. "have become" is inaccurate and unclear as it connotes time.

Line 41. Please consider changing "contaminated" to colonized, as legionella are normal inhabitants of drinking water, not contaminants.

Line 43. By "bubble baths" is meant spas and hot tubs?

Line 62. What is meant by "renovated water supply systems"

Lines 145-148 and Table 2. Please consider using correlation as the statistical analysis tool.

Lines 173-178. This analysis does address one of the variables influencing studies of co-existence of legionella and amoebae. Accordingly, please consider describing the basis for the Chi-Square analysis (e.g., what was the null hypothesis and expected group)? Please consider employ correlation between detection (numbers as well) of a specific amoebae genus and legionella. Please ensure that the analysis was not biased by inclusion of samples with more than one amoebae genus..

Lines 233-244. I assume that reporting Legionella pneumophila infections is not required in Latvia. If so, please include that information..

Lines 266-251. Unnecessary.

Author Response

Reviewer 1

Manuscript has been revised by authors and improvements were made in following sections: Abstract, Introduction, Materials and Methods, Results, Discussion and Conclusions. Recommendations of Reviewers were incorporated in the text, including typing and grammar errors, correct wording etc.

All changes are marked using track changes option in MS Word.

Specific comment

Line 14. "have become" is inaccurate and unclear as it connotes time.

Manuscript text changed

Line 41. Please consider changing "contaminated" to colonized, as legionella are normal inhabitants of drinking water, not contaminants.

Manuscript text changed

Line 43. By "bubble baths" is meant spas and hot tubs?

Manuscript text changed

Line 62. What is meant by "renovated water supply systems"

Author replay: Renovation in our understanding means pipelines were replaced

Lines 145-148 and Table 2. Please consider using correlation as the statistical analysis tool.

Manuscript text changed

Lines 173-178. This analysis does address one of the variables influencing studies of co-existence of legionella and amoebae. Accordingly, please consider describing the basis for the Chi-Square analysis (e.g., what was the null hypothesis and expected group)? Please consider employ correlation between detection (numbers as well) of a specific amoebae genus and legionella. Please ensure that the analysis was not biased by inclusion of samples with more than one amoebae genus.

Manuscript text changed

Lines 233-244. I assume that reporting Legionella pneumophila infections is not required in Latvia. If so, please include that information.

Manuscript text changed

Lines 266-251. Unnecessary.

Author replay: Included in the Journal Template and stated as mandatory

Reviewer 2 Report

Authors must review the language and specify the acronym FLA in the text. I believe it is necessary to deepen introduction and discussions.

Author Response

Reviewer 2

Manuscript has been revised by authors and improvements were made in following sections: Abstract, Introduction, Materials and Methods, Results, Discussion and Conclusions. Recommendations of Reviewers were incorporated in the text, including typing and grammar errors, correct wording etc.

All changes are marked using track changes option in MS Word.

Specific comment

Authors must review the language and specify the acronym FLA in the text.

Manuscript text changed

I believe it is necessary to deepen introduction and discussions.

Minor changes were made in Introduction and Discussion sections

Reviewer 3 Report

The manuscript entitled „Co-occurrence of free-living amoeba and Legionella in drinking water supply system” by Valcina et al. is an interesting study about the occurrence and prevalence of Legionella and FLA in Latvian water systems. The study provides a good overview about the presence of both microorganisms and the concentration of Legionella in the country focusing in Riga, the capital of the country. Moreover, authors also reported the diversity of Legionella and FLA on the study area and studied if there is any correlation between the presence of Legionella and FLA and several environmental parameters such as water temperature or source of the water supply.

In general it is a well written manuscript very easy to read. There are some concerns I would like to discuss with the authors. Find them below point by point:

Title: „Co-occurrence of free-living amoeba and Legionella in drinking water supply systems” authors can also write drinking water systems. The term is generally used like that too. If they change it, please change it consistently throughout the manuscript. Line 13-14. What was first, the egg of the chicken? In this case, FLA were first and they applied a selective pressure on water bacteria to avoid amoeba digestion. So, Legionella evolved to avoid amoeba digestion and turned predators into hosts. Please, re-write the sentence similar to Line 54. The correct name is groundwater (instead of underground water), please change it consistently throughout the manuscript. Line 27 (also Discussion Line 244), authors didn’t check the presence of VBNC pathogens. I would not mention it here. Moreover, the health-relevance of VBNC Legionella for example is still under discussion. Line 29-31 (also Conclusions Line 248-251). Authors say that traditional monitoring methods may not be sufficient for Legionella control because of the high incidence of legionella-related FLA. What do authors mean with this sentence? Do authors suggest to include FLA in the surveillance programs?

Results of the present study showed that FLA occurred more often in drinking water systems than Legionella. When Legionella was detected, so were FLA but when FLA were observed Legionella was not always detected.

Please define a bit better what do you mean with the sentence or may be soften it.

Line 29-31 (also Conclusions Line 248-251). I would use risk management programs/plans instead of activities. Line 36. Found instead of met Line 39. Pontiac fever is not only underreported because it is not a notifiable disease but also because is very rarely diagnosed. Doctors cannot report what they do not see/test. Line 51. Predators for other microorganisms Line 53. Man-made water systems. Line 56. Such as nutrient source and additional protection against temp…. Line 57. I would change the word ambushes for e.g. escape the grazing effect, predation etc. Line 58. Legionella (italic? Check consistency when writing Legionella) Line 58-59. Check the English in the last part of the sentence (after the citations 10,11,12) Line 62. The buildings. I would recommend a minor English check throughout the manuscript. Line 82. Which kind of filters did the authors use? Polycarbonate? Please, mention it Was the concentrate on the filters somehow resuspended before transfer it to BCYE agar? Please, specify. Specify that authors used the ISO 11731:1998 or 2017. Line 92-94. Without the rice grains, PAS is more a buffer than a media because there are no nutrients. How did authors prepare the final PAS buffer? How much volume of the two solutions? Line 96. Describe the PYG medium. Line 101. PCR was done using a protocol previously described (citation), use the same sentence format for Lines 103 and 108. What are ASA.S1 and DF3, please provide more info.Line 110. Kit was used. Line 114. Query coverage? Average? Line 121. Legionella spp. were detected IN 114…..check throughout the manuscript. Line 130. According to the methodology described, the lowest concentration of Legionella that authors could detect is 50 cfu/L. (Filter 1 L, resuspend filter in 5 mL, from there they transfer to the plate 0.1 mL, that means that if they found 1 colony has to be multiplied by 50 to have the results in cfu/L). For acid treatment, if they add 0.1 mL of the concentrate into 0.9 mL of acid and they transferred the whole 1 mL to the plate is the same limit of detection, if they transferred 0.1 mL the limit of detection is 500 cfu/L). Please, check your results and correct accordingly! Line 141. What was the average temperature of cold water? I would suggest authors to group results by microorganisms. Thus, I would move paragraphs from Line 182-183 and Table 6 to Line 142. And then I would place the information about the co-occurrence of microorganisms Line 142-144 and Table 2 in Line 169. Line 143. No Legionella positive samples (cases can mislead to infection cases). Table 3. Data shown belong to microscopic observations or to PCR results, please specify. I’m curious, what happens with the occurrence of FLA in waters >55°C? Was the FLA diversity in that case also higher or only for hot water samples <55°C? Did authors take different samples from the same building? When yes, was there any difference between Legionella and FLA occurrence or species serogroup diversity? Line 205. With a study carried out in Hungary (add citation at the end of the sentence) Line 229. If the temperature of the hot water system is already low (<50°C) both proximal and distal areas of the system have an increased health risk. How did authors measure the temperature of the system? Did they wait to until the water reached a certain temperature or took the sample without flushing? Please, specify. Line 241. What do authors mean with the last part of the sentence? It is still to be proven that Legionella detected by PCR or VBNC suppose and elevated health risk in comparison to the culturable ones. In a recent study, Cervero-Arago et al. 2019 showed that culturable Legionella were precisely the most virulent. Please, consider changing the sentence. Line 242. Authors did not check for the persistence of Legionella in the different buildings. Also did not mention in which time-frame samples were taken. If they did, meaning that they took samples from the same sampling points repeatedly, please indicate that in the M&M section. Line 244. I don’t see how the results of the study may suggest that VBNC pathogens have to be considered in new water safety approached. Please, correct. Conclusions (already mentioned it in the abstract section).

Author Response

Reviewer 3

Manuscript has been revised by authors and improvements were made in following sections: Abstract, Introduction, Materials and Methods, Results, Discussion and Conclusions. Recommendations of Reviewers were incorporated in the text, including typing and grammar errors, correct wording etc.

All changes are marked using track changes option in MS Word.

Specific comment

Title: „Co-occurrence of free-living amoeba and Legionella in drinking water supply systems” authors can also write drinking water systems. The term is generally used like that too. If they change it, please change it consistently throughout the manuscript.

Authors have decided not to change the title of the manuscript.

Line 13-14. What was first, the egg of the chicken? In this case, FLA were first and they applied a selective pressure on water bacteria to avoid amoeba digestion. So, Legionella evolved to avoid amoeba digestion and turned predators into hosts. Please, re-write the sentence similar to

Manuscript text changed

Line 54. The correct name is groundwater (instead of underground water), please change it consistently throughout the manuscript.

Manuscript text changed

Line 27 (also Discussion Line 244), authors didn’t check the presence of VBNC pathogens. I would not mention it here. Moreover, the health-relevance of VBNC Legionella for example is still under discussion.

Authors have decided to exclude related statements from Discussion and Conclusions sections.

Line 29-31 (also Conclusions Line 248-251). Authors say that traditional monitoring methods may not be sufficient for Legionella control because of the high incidence of legionella-related FLA. What do authors mean with this sentence? Do authors suggest to include FLA in the surveillance programs? Results of the present study showed that FLA occurred more often in drinking water systems than Legionella. When Legionella was detected, so were FLA but when FLA were observed Legionella was not always detected. Please define a bit better what do you mean with the sentence or may be soften it.

Manuscript text changed

Line 29-31 (also Conclusions Line 248-251). I would use risk management programs/plans instead of activities.

Manuscript text changed

Line 36. Found instead of met

Manuscript text changed

Line 39. Pontiac fever is not only underreported because it is not a notifiable disease but also because is very rarely diagnosed. Doctors cannot report what they do not see/test.

Manuscript text changed

Line 51. Predators for other microorganisms

Manuscript text changed

Line 53. Man-made water systems.

Manuscript text changed

Line 56. Such as nutrient source and additional protection against temp….

Manuscript text changed

Line 57. I would change the word ambushes for e.g. escape the grazing effect, predation etc.

Manuscript text changed

Line 58. Legionella (italic? Check consistency when writing Legionella)

Manuscript text changed

Line 58-59. Check the English in the last part of the sentence (after the citations 10,11,12)

Manuscript text changed

Line 62. The buildings. I would recommend a minor English check throughout the manuscript.

Minor English check were made

Line 82. Which kind of filters did the authors use? Polycarbonate? Please, mention it Was the concentrate on the filters somehow resuspended before transfer it to BCYE agar? Please, specify. Specify that authors used the ISO 11731:1998 or 2017.

Information provided in the Materials and Methods section

Line 92-94. Without the rice grains, PAS is more a buffer than a media because there are no nutrients. How did authors prepare the final PAS buffer? How much volume of the two solutions?

Information provided in the Materials and Methods section

 Line 96. Describe the PYG medium.

Information provided in the Materials and Methods section

Line 101. PCR was done using a protocol previously described (citation), use the same sentence format for

Manuscript text changed

Lines 103 and 108. What are ASA.S1 and DF3, please provide more info.

Information provided in the Materials and Methods section

Line 110. Kit was used. Line 114. Query coverage? Average?

Typing error corrected

 Line 121. Legionella spp. were detected IN 114…..check throughout the manuscript.

Typing error corrected

Line 130. According to the methodology described, the lowest concentration of Legionella that authors could detect is 50 cfu/L. (Filter 1 L, resuspend filter in 5 mL, from there they transfer to the plate 0.1 mL, that means that if they found 1 colony has to be multiplied by 50 to have the results in cfu/L). For acid treatment, if they add 0.1 mL of the concentrate into 0.9 mL of acid and they transferred the whole 1 mL to the plate is the same limit of detection, if they transferred 0.1 mL the limit of detection is 500 cfu/L). Please, check your results and correct accordingly!

Typing error corrected

Line 141. What was the average temperature of cold water? I would suggest authors to group results by microorganisms. Thus, I would move paragraphs from Line 182-183 and Table 6 to Line 142. And then I would place the information about the co-occurrence of microorganisms Line 142-144 and Table 2 in Line 169.

Information added and Results section rearranged

Line 143. No Legionella positive samples (cases can mislead to infection cases).

Manuscript text changed

Table 3. Data shown belong to microscopic observations or to PCR results, please specify.

Information specified

I’m curious, what happens with the occurrence of FLA in waters >55°C? Was the FLA diversity in that case also higher or only for hot water samples <55°C? Did authors take different samples from the same building? When yes, was there any difference between Legionella and FLA occurrence or species serogroup diversity?

Information in lines 167 – 169, added lines 210 - 211

Line 205. With a study carried out in Hungary (add citation at the end of the sentence)

Manuscript text changed

Line 229. If the temperature of the hot water system is already low (<50°C) both proximal and distal areas of the system have an increased health risk. How did authors measure the temperature of the system? Did they wait to until the water reached a certain temperature or took the sample without flushing? Please, specify.

Information provided in the Materials and Methods section.

Line 241. What do authors mean with the last part of the sentence? It is still to be proven that Legionella detected by PCR or VBNC suppose and elevated health risk in comparison to the culturable ones. In a recent study, Cervero-Arago et al. 2019 showed that culturable Legionella were precisely the most virulent. Please, consider changing the sentence.

Authors have decided to exclude related statements from Discussion and Conclusions sections.

Line 242. Authors did not check for the persistence of Legionella in the different buildings. Also did not mention in which time-frame samples were taken. If they did, meaning that they took samples from the same sampling points repeatedly, please indicate that in the M&M section.

Information provided in the Materials and Methods section.

Line 244. I don’t see how the results of the study may suggest that VBNC pathogens have to be considered in new water safety approached. Please, correct. Conclusions (already mentioned it in the abstract section).

Authors have decided to exclude related statements from Discussion and Conclusions sections